# Transient birefringence of liquids induced by terahertz electric-field torque on permanent molecular dipoles

Mohsen Sajadi[1], Martin Wolf[1] & Tobias Kampfrath[1]

Collective low-frequency molecular motions have large impact on chemical reactions and structural relaxation in liquids. So far, these modes have mostly been accessed indirectly by off-resonant optical pulses. Here, we provide evidence that intense terahertz (THz) pulses can resonantly excite reorientational-librational modes of aprotic and strongly polar liquids through coupling to the permanent molecular dipole moments. We observe a significantly enhanced response because the transient optical birefringence is up to an order of magnitude higher than obtained with optical excitation. Frequency-dependent measurements and a simple analytical model indicate that the enhancement arises from resonantly driven librations and their coupling to reorientational motion, assisted by the pump field and/or a cage translational mode. Our results open up the path to applications such as efficient molecular alignment, enhanced transient Kerr signals and systematic resonant nonlinear THz spectroscopy of the coupling between intermolecular modes in liquids.

[1] Department of Physical Chemistry, Fritz Haber Institute of the Max Planck Society, 14195 Berlin, Germany. Correspondence and requests for materials should be addressed to M.S. (email: sajadi@fhi-berlin.mpg.de).

L ow-frequency structural dynamics of liquids in the range from 0.1 to 10 THz (3 to 330 cm$^{-1}$) are believed to strongly contribute to the outcome of chemical processes[1–5]. The underlying molecular motions can be complex and include reorientations, vibrations and translations. To access and trigger rotational dynamics, one may take advantage of the torque

$$\mathbf{T} = (\boldsymbol{\mu}_0 + \boldsymbol{\mu}_{ind}) \times \mathbf{E} \qquad (1)$$

exerted on molecules by an external time ($t$)-dependent electric field $\mathbf{E}(t)$. Coupling is mediated by two types of electric dipoles: the permanent molecular dipole moment $\boldsymbol{\mu}_0(t)$ (having constant modulus $\mu_0$) and the instantaneous dipole moment $\boldsymbol{\mu}_{ind}(t)$ induced by polarizing the molecule's electron distribution.

For linearly polarized $\mathbf{E}$ and molecules with cylindrical symmetry, the torque due to $\boldsymbol{\mu}_{ind}$ scales with $\Delta\alpha \mathbf{E}^2(t)$, where $\Delta\alpha$ quantifies the difference of the polarizability parallel and perpendicular to the molecular axis[6]. Since squaring of $\mathbf{E}$ rectifies the rapidly oscillating light field, femtosecond laser pulses are routinely used to exert ultrafast torque of type $\boldsymbol{\mu}_{ind} \times \mathbf{E}$ on solvent molecules. In contrast, optical pulses yield a vanishing time-integrated torque of type $\boldsymbol{\mu}_0 \times \mathbf{E}$ because $\mathbf{E}$ changes the direction of the permanent dipole $\boldsymbol{\mu}_0$ very little over the only ∼1 fs long optical half-cycle[7,8]. Therefore, to act on permanent dipole moments, we need to abandon the rapid field oscillations inherent to optical stimuli and instead make use of sup-picosecond transient electric fields.

In recent ultrafast works[9–12], such intense terahertz (THz) pulses were already successfully employed to drive liquids, while the resulting dynamics were traced by detecting the transient optical birefringence. In addition to an instantaneous electronic response, longer-lived signals were found and assigned to reorientational molecular motions[9,11] and intramolecular vibrational modes[11]. By using two pump pulses with variable delay, Finnegan et al.[12] were even able to reveal anharmonic coupling between THz vibrational modes. These pioneering studies suggest that THz-pump optical-birefringence spectroscopy is a highly promising tool to also gain insights into the role of permanent and induced molecular electric dipoles during the interaction with intense sub-picosecond electric fields. Indeed, recent studies on gases suggest that THz field torque on molecules with permanent electrical dipole moment should have a considerable impact on the nonlinear THz response of liquids[13].

In this article, we make use of intense THz pulses to exert ultrafast torque on polar and nonpolar solvent molecules. For strongly polar liquids, such as dimethyl sulfoxide (DMSO), we find that the pump-induced transient optical birefringence is enhanced by more than one order magnitude as compared to optical excitation. This enhancement highlights the significance of THz field coupling to the permanent molecular dipole moments. Our experimental observations are consistent with a simple model of the driven liquid, pointing to a resonantly enhanced process and to coupling of librational and reorientational modes. Thus, resonant nonlinear excitation of liquids by THz fields bears great potential for applications such as efficient molecular alignment, enhanced transient Kerr signals and systematic studies of the coupling of intermolecular modes in liquids.

## Results

### Experiment.
A schematic of our experiment is shown in Fig. 1a; details are described in the 'Methods' section. An intense, linearly polarized and phase-locked THz electromagnetic pulse (see amplitude spectra in Fig. 1b) is incident onto a polar liquid consisting of axially symmetric molecules. The resulting transient birefringence (THz Kerr effect, TKE)[9] is monitored by a time-delayed optical probe pulse whose polarization acquires an elliptical polarization. The degree of ellipticity scales with the

difference $\Delta n = n_{\parallel} - n_{\perp}$ between the liquid's optical refractive index parallel ($n_{\parallel}$) and perpendicular ($n_{\perp}$) to the driving field $\mathbf{E}$ (refs 14,15). The $\Delta n$, in turn, is proportional to the ensemble average[6,13]

$$\Delta n(t) \propto \int du\, \Delta\alpha\, f(u,t) P_2(u), \qquad (2)$$

where $u = \cos\theta$ is the cosine of the angle between $\mathbf{E}$ and the molecular axis (Fig. 1a), $f(u,t)$ is its instantaneous population distribution and $P_2(u)$ is proportional to $u^2 - 1/3$. To compare the anisotropy induced by THz and optical excitation, we conduct the same measurements but with the THz pulse replaced by an optical pump pulse (optical Kerr effect, OKE). The instantaneous intensity of optical and THz pump pulse have approximately identical shape (see inset of Fig. 2a), thereby allowing for straightforward comparison of TKE and OKE data.

In our study, we focus on simple polar liquids, DMSO and chloroform, for two reasons. First, their coupling to the incident THz field is predominantly mediated by the molecular inclination angle $\theta$ (Fig. 1a). Direct coupling to intramolecular degrees of freedom is expected to be negligible at the frequencies below 3 THz considered here. This notion is supported by the THz absorption spectra of the liquids, which are shown in Fig. 1b. For chloroform, the amplitude spectrum of the THz pump (LN source) overlaps with both reorientational (at ∼0.2 THz) and librational modes (∼1 THz)[16]. For DMSO, the reorientational mode is at much lower frequencies (∼10 GHz)[17], and the THz pump spectra instead overlap with a librational mode[18].

Second, the liquids chosen here allow us to systematically compare the torque induced by coupling to induced electronic and to permanent dipoles because they exhibit distinctly different magnitude combinations of $\Delta\alpha$ and $\mu_0$: DMSO ($\mu_0 \approx 4.1$ D and $\Delta\alpha > 0$) and chloroform ($\mu_0 \approx 1.12$ D and $\Delta\alpha < 0$)[19]. To calibrate our comparative procedure, we perform OKE and TKE experiments on the nonpolar liquids toluene and cyclohexane ($\mu_0 \approx 0$).

### Transient birefringence.
The transient birefringence of three liquids following THz and optical excitation is shown in Fig. 2. For comparison, the squared THz pump field $\mathbf{E}^2(t)$ and instantaneous optical pump intensity $I(t)$ are plotted in the inset of Fig. 2a. The signal amplitude is found to grow linearly with the pump power, for both THz and optical excitation (see Supplementary Fig. 1). Note that all TKE and OKE signals share two common features: (i) a sharp initial rise with a shape similar to the squared THz pump field $\mathbf{E}^2(t)$ and optical intensity envelope $I(t)$, respectively (inset of Fig. 2a), followed by (ii) a slower decay on a picosecond time scale.

Feature (i) is assigned to the response of the electronic subsystem of the sample[9,14,15]. This response is instantaneous because in our experiment, the excitation energies ($>5$ eV) of the electrons are much larger than the photon energies of the THz ($<10$ meV) and optical pump pulse (∼1.5 eV).

Once pump and probe pulses do not overlap any more, the dynamics are dominated by the slower feature (ii), which is assigned to the relaxation of the nuclear degrees of freedom of the molecules. Figure 2a and the insets in Fig. 2b,c reveal a remarkable observation for all three liquids: at pump-probe delays $t > 1$ ps, the dynamics are independent of the pump pulse used (THz or optical), apart from a global signal scaling factor. This finding shows that all the modes we observe in our experiment can be driven by both THz and optical pump pulses. The mono-exponential birefringence decay of DMSO (time constant of ∼6.4 ps) and the bi-exponential signal decay of chloroform (time constants ∼0.4 and 2 ps) are in line with

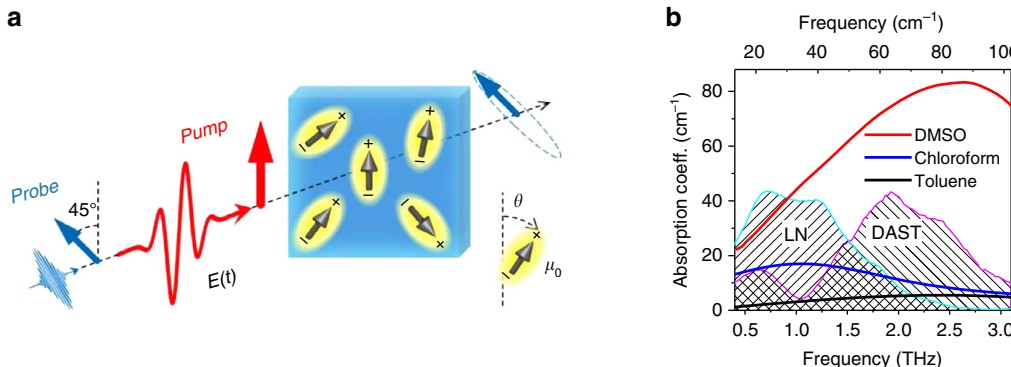

**Figure 1 | TKE and OKE in dipolar liquids.** (**a**) An intense THz or optical pump pulse induces birefringence in a polar liquid. The transient birefringence is measured by an optical probe pulse that becomes elliptically polarized on propagation through the medium. We study liquids with various values of the permanent molecular dipole moment $\mu_0 = |\mathbf{\mu}_0|$ and polarizability anisotropy $\Delta\alpha$. (**b**) Equilibrium THz absorption spectra of DMSO, chloroform and toluene. Amplitude spectra of THz pump pulses from two different sources (LN and DAST, see 'Methods' section) are shown by dashed areas.

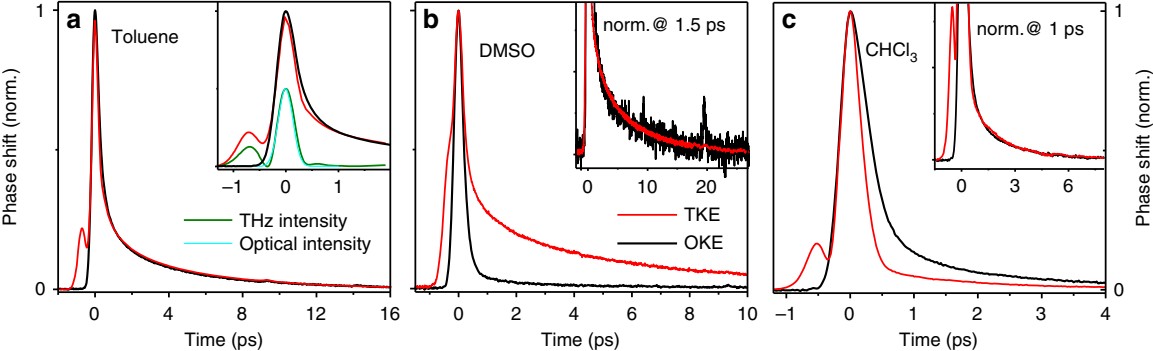

**Figure 2 | Transient optical birefringence of liquids following THz and optical excitation.** (**a**) TKE (red line) and OKE (black line) signals of toluene. Signals are normalized to the initial peak signal where the instantaneous electronic contribution is expected to dominate the birefringence signal. The inset shows the instantaneous intensity of the THz pump pulse ($\mathbf{E}^2(t)$, green line) and the optical pump pulse ($I(t)$, cyan line). (**b,c**) Same as **a**, but with data taken on DMSO (**b**) and chloroform (**c**), respectively. For DMSO, the maximum phase shift amounts to 6 mrad at an incident peak THz field of 2 MV cm$^{-1}$. The insets in **b,c** show the main-panel data but normalized to the signal amplitude at a delay of 1.5 ps and 1 ps, respectively.

previous OKE studies[20,21], in which the slower time constants were assigned to reorientational relaxation.

To evaluate how efficiently these modes are excited by the THz and optical pump pulse, we normalize TKE and OKE signals to their respective peak value found around $t = 0$, as has already been done for the curves in the main panels of Fig. 2. As detailed in the 'Methods' section, this procedure is tantamount to normalizing the signals to the pump intensity. Therefore, once pump–probe overlap is gone ($t > 1$ ps), normalized signal amplitudes approximately equal the relative strength with which THz and optical pump pulses drive the nuclear dynamics.

**Nonpolar versus polar liquids.** Interestingly, for toluene (Fig. 2a), identical normalized dynamic birefringence for THz and optical excitation are found at delays larger than the pump duration. Such agreement indicates that both THz and optical pump field couple to the rotational degrees of freedom with the same strength, consistent with our expectation: due to the relatively small permanent molecular dipole moment $\mathbf{\mu}_0$ of toluene, torque is dominated by the pump-induced moment $\mathbf{\mu}_{\text{ind}}$ (see equation (1)). As both the THz and optical pump photon energies are far off any electronic or vibrational resonance and since THz ($\mathbf{E}^2(t)$) and optical ($I(t)$) pump pulse have approximately identical shape (inset of Fig. 2a), both pulses exert identical normalized torques $\mathbf{\mu}_{\text{ind}} \times \mathbf{E}$ on molecules. Our

interpretation is confirmed by measurements on another nonpolar liquid, cyclohexane, where we also observe identical normalized dynamics following THz and optical excitation (see Supplementary Figs 2 and 3 and Supplementary Note 1). This observation also implies that the weak THz pump absorption by transient electric dipoles, for instance arising from collisions[22], makes a negligible signal contribution within the accuracy of our normalization procedure.

Remarkably and in stark contrast to the nonpolar liquid toluene, the normalized amplitude of the nuclear relaxation signal of the polar liquids DMSO (Fig. 2b) and chloroform (Fig. 2c) is seen to depend strongly on the pump frequency. While THz excitation of DMSO yields a ~10 times larger normalized birefringence signal than an optical pump (Fig. 2b), a signal reduction by a factor of ~3 is observed for chloroform (Fig. 2c). This observation indicates that in DMSO, the THz field couples more strongly to the rotational degrees of freedom than an optical pulse, whereas a reversed situation is found for chloroform. Note that this variation of coupling strength is strongly correlated with the material constants $\Delta\alpha$ and $\mu_0$. For $\mu_0 = 0$, identical birefringence is observed for THz and optical excitation (Fig. 2a). In polar liquids ($\mu_0 > 0$), however, THz-induced birefringence is enhanced for $\Delta\alpha > 0$ (Fig. 2b) but reduced for $\Delta\alpha < 0$ (Fig. 2c) as compared to optical excitation. This trend is supported by data on four other liquids (see Supplementary Figs 3 and 4).

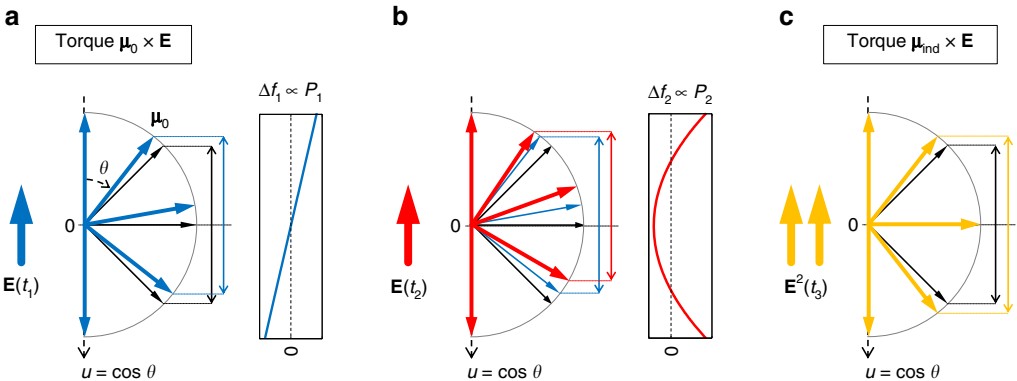

**Figure 3 | Model of transient birefringence due to ultrafast torque.** (**a**) In equilibrium, the orientation of permanent molecular dipoles is isotropic (black arrows), with a distribution function $f = f_0$ independent of the molecular inclination angle $\theta$. At time $t_1$, a field kick $\mathbf{E}(t_1)$ exerts torque of type $\boldsymbol{\mu}_0 \times \mathbf{E}$, which orients molecules along $\mathbf{E}$. This torque induces a $P_1$-type change $\Delta f_1$ in the distribution function and a dielectric polarization. Note that pairs $(\theta, 180° - \theta)$ of molecules are rotated rigidly. Therefore, the sum of the arrow lengths projected on the $u$ axis remains unchanged (see black and blue double arrows), and no optical birefringence is induced. (**b**) At time $t_2 > t_1$, a second field kick triggers additional rotation, but in contrast to panel **a**, dipoles in the lower hemisphere experience more torque than in the upper hemisphere. As a consequence, the change $\Delta f_2$ in the distribution function is $P_2$-like and accompanied by optical birefringence, as can be seen from the modified projected arrow lengths (blue and red double arrows) and from a quantitative model (see 'Methods' section). (**c**) Torque of type $\boldsymbol{\mu}_{\text{ind}} \times \mathbf{E}$ at time $t_3$ scales with $\mathbf{E}^2(t_3)$ and rotates each molecule to the closest pole (orange arrows), resulting in a $P_2$-type change in $f$ (see **b**) and optical birefringence (see black and orange double arrows).

**Model**. To obtain an interpretation of the measured birefringence dynamics $\Delta n(t)$ (Fig. 2), it is instructive to consider the impact of a THz or optical pump pulse on the angular distribution function $f(u,t)$ of the solvent molecules (Fig. 3). Since $\Delta n(t)$ is found to scale linearly with the pump power (Supplementary Fig. 1), two interactions with the pump field are required. In equilibrium, the distribution function $f = f_0$ is isotropic and independent of the molecular inclination angle $\theta$ (Fig. 3a). When a $\delta$-like electric-field pulse is incident at time $t = t_1$, it exerts a torque through the permanent molecular dipole moment $\boldsymbol{\mu}_0$. Owing to equations (1) and (9), a perturbation $\Delta f_1(u,t)$ of the isotropic $f_0$ is induced that follows $P_1(u) = u = \cos \theta$ (Fig. 3a). The $P_1$-type dependence implies that molecules at $\theta$ and $180° - \theta$ rotate in phase. Therefore, $\Delta f_1$ does not yet cause optical birefringence but is accompanied by a time-dependent dielectric polarization $\mathbf{P} = \chi^{\mu_0} \mathbf{E}$. In this convolution, $\chi^{\mu_0}(t)$ is the $\boldsymbol{\mu}_0 \times \mathbf{E}$-type contribution to the dielectric susceptibility[23] that describes the temporal build-up and decay of $\mathbf{P}(t)$.

On interaction with a second $\delta$-like field pulse at time $t = t_2$, part of $\Delta f_1$ is converted into a new distribution component that scales with the two fields at $t = t_1$ and $t_2$ and whose shape follows $P_2(u) = (3u^2 - 1)/2$ (see Fig. 3b and equation (10)). The $P_2$-type dependence can be understood as arising from a motion during which molecules at $\theta$ and $180° - \theta$ rotate out of phase.

A $P_2$-like modification also results from a single perturbation by the torque $\boldsymbol{\mu}_{\text{ind}} \times \mathbf{E} \propto \Delta\alpha \mathbf{E}^2$ related to the induced electronic dipole moment $\boldsymbol{\mu}_{\text{ind}}$ (see Fig. 3c). The total $P_2$-type perturbation $\Delta f_2(u,t)$ due to the two torques causes transient optical birefringence that is measured in our experiment (equation (2)). By developing a simple but relatively general model[24–27] for the dynamics of $f(u,t)$ (see 'Methods' section), we derive the transient optical birefringence.

$$\Delta n(t) \propto R_2 [E \cdot (N\Delta\alpha E + 3\chi^{\mu_0} E)]. \qquad (3)$$

Here, $E(t)$ is the amplitude of the linearly polarized optical or THz pump field, $N$ is the number of molecules per volume, and $\chi^{\mu_0}$ is the contribution of the permanent electric dipole moment $\boldsymbol{\mu}_0$ to the familiar total dielectric susceptibility (see above).

Note that equation (3) reveals an analogy of the $\boldsymbol{\mu}_{\text{ind}}$- and $\boldsymbol{\mu}_0$-related coupling mechanisms: the first field interaction generates an effective electronic ($N\Delta\alpha E$) and orientational polarization ($\chi^{\mu_0} E$) which, in turn, serves as a handle for the second field interaction to generate a $P_2$-like perturbation (square bracket in equation (3)). The decay of the resulting $P_2$-type change in the angular distribution function is captured by the response function $R_2(t)$, which is independent of the way the $P_2$-modification was generated (see equation (10)).

We note that equation (3) is consistent with our central experimental findings: first, in all liquids studied, we observe identical relaxation dynamics for both optical and THz excitation (see Fig. 2a and insets in Fig. 2b,c). This agreement suggests that the picosecond decay of the optical birefringence is a manifestation of the $P_2$-relaxation function $R_2(t)$. Second, for $\mu_0 = 0$ and thus $\chi^{\mu_0} = 0$, no enhancement of birefringence is expected based on equation (3), in agreement with the experimental result (Fig. 2a). Third, equation (3) implies that for THz versus optical pumping, the normalized $\Delta n(t)$ is enhanced by a factor that scales with $1 + 3\chi^{\mu_0}/N\Delta\alpha \propto 1 + B\mu_0^2/\Delta\alpha$, where $B$ is a positive constant (see 'Methods' section). Thus, THz excitation should yield larger or smaller birefringence than optical excitation when polar liquids with, respectively, positive or negative $\Delta\alpha$ are used, again in agreement with our experimental findings (Fig. 2b,c).

## Discussion

According to equation (3), the THz-induced optical birefringence depends critically on $\chi^{\mu_0}$, the $\boldsymbol{\mu}_0 \times \mathbf{E}$-related component of the dielectric susceptibility. Various modes of diffusive reorientation[23] (Debye modes) as well as hindered rotations[28] (librations) contribute to $\chi^{\mu_0}$. The dielectric response of DMSO is dominated by a Debye-type component[17], which peaks at $\sim 10$ GHz. If this mode were the only contribution to $\chi^{\mu_0}$, THz excitation would be virtually off-resonant and result in reduced transient optical birefringence with increasing pump frequency (see Supplementary Fig. 5 and Supplementary Note 2).

Note, however, the opposite trend is observed in the experiment, as seen in Fig. 4a: when the centre frequency of the THz pump is shifted from $\sim 1$ to $\sim 2$ THz, the birefringence signal increases notably, thereby suggesting the presence of a resonance[18]. We assign this observation to resonant excitation of the librational mode of DMSO, which is considered to be the origin of the broad absorption feature[18] at 2.5 THz (Fig. 1b). This

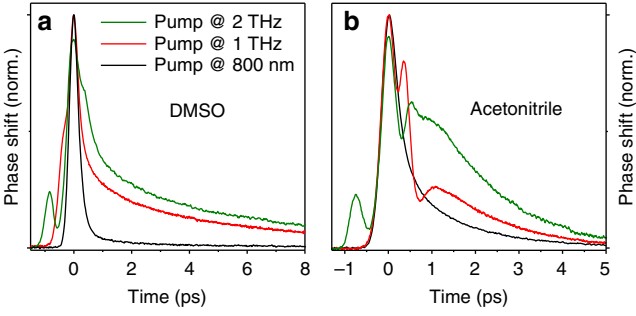

**Figure 4 | Impact of the THz pump frequency.** (**a**) Normalized transient birefringence signals of DMSO following excitation with THz pump pulses centred at $\sim 1\,\mathrm{THz}$ (red line) and $\sim 2\,\mathrm{THz}$ (green line). The THz pump spectra are shown in Fig. 1b (for THz temporal waveforms see Supplementary Fig. 7). For comparison, the signal induced by optical excitation is also shown (black line). (**b**) Same as **a**, but for acetonitrile.

assignment is bolstered by repeating the experiment on another dipolar liquid, acetonitrile, which exhibits a Debye mode at $\sim 50\,\mathrm{GHz}$ and a librational resonance at $\sim 3\,\mathrm{THz}$ (see Supplementary Fig. 6)[29]. As seen in Fig. 4b, we observe an increased birefringence when the pump centre frequency is increased from $\sim 1$ to $\sim 2\,\mathrm{THz}$.

The results of Fig. 4 indicate that resonant excitation of the librational mode leads to reorientational molecular motion with enhanced amplitude. This observation and our transient birefringence model (Fig. 3 and equation (3)) suggest the excitation scheme depicted in Fig. 5a: the $\mu_0 \times \mathbf{E}$ torque of the first interaction with the pump field (arrow 1) resonantly drives hindered molecular rotation with $P_1$-like angular distribution (Lib($P_1$)). This motion plus the second field interaction cause an impulsive $P_2$-type perturbation that induces both $P_2$-distributed librational (see arrow 2a and Lib($P_2$)) and reorientational dynamics (arrow 2b and Reo($P_2$)). Figure 5a shows that the transition paths 1 plus 2a and 1 plus 2b can be considered as stimulated processes, resonantly enhanced by the intermediate librational mode Lib($P_1$).

While in path 2a, the coupling of librational (Lib($P_1$)) and reorientational (Reo($P_2$)) motion is assisted by the pump field, Reo($P_2$) can also be excited by conversion of Lib($P_2$) into Reo($P_2$) (see arrow 3 in Fig. 5a). According to a model of Fayer and co-workers[30], this process is assisted by so-called translational $\beta$ modes[31–36], which lead to temporal changes of the cage surrounding the librating molecule. As a consequence, this molecule does not relax to its initial direction, thereby resulting in net alignment and, thus, orientational diffusion (Reo($P_2$)) with increased amplitude. The efficiency of this coupling process (arrow 3) is the larger, the closer the time scales of librational damping and $\beta$ mode are[30]. Interestingly, the $\beta$ mode of acetonitrile (at $0.5\,\mathrm{THz}$) has larger overlap with the libration than that of DMSO. This fact could contribute to the pronounced increase of the birefringence signal of acetonitrile when the pump spectrum is shifted toward the libration resonance.

Currently, the librational dynamics are faster than the time resolution of our TKE set-up, but extension by a second time-delayed pump pulse[12] should make observation of the Lib($P_2$) mode possible. Along similar lines, replacing the off-resonant optical probe pulse by a resonant THz probe should allow us to shed more light on the relaxation process along path 3.

Note that the resonant excitation of the libration (arrow 1 in Fig. 5a) is only possible because the THz field couples to the

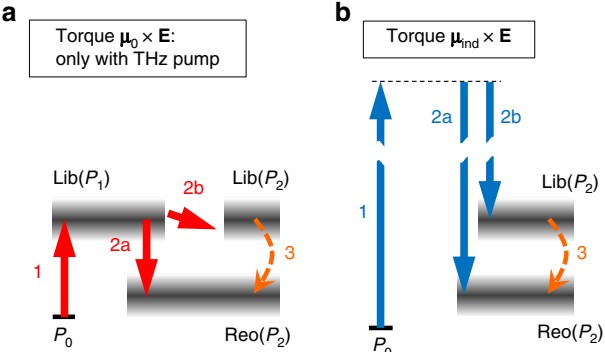

**Figure 5 | Schematics of pump-driven transitions and relaxation pathways.** (**a**) For THz excitation through $\mu_0 \times \mathbf{E}$-type torque, the first field interaction (arrow 1) resonantly excites a $P_1$-distributed libration Lib($P_1$). A second interaction with the THz pump field causes a $P_2$-like perturbation and, thus, transition to $P_2$-distributed reorientational motion (arrow 2a) and libration (arrow 2b). Due to mode coupling, the libration is partially transformed into reorientational motion[30] (arrow 3), thereby increasing the amplitude of the reorientational mode. Note that all excited levels are strongly broadened. (**b**) Optical excitation induces off-resonant $\mu_{\mathrm{ind}} \times \mathbf{E}$-type torque scaling with $\mathbf{E}^2$ that drives $P_2$-distributed reorientational and librational motion. Analogous to panel **a**, Lib($P_2$) may be converted into Reo($P_2$) by mode coupling.

permanent molecular dipole moment. In contrast, for optical excitation, the $\mu_{\mathrm{ind}} \times \mathbf{E}$ torque dominates, and the Raman process shown in Fig. 5b prevails[7]. The first field interaction (arrow 1) is off-resonant because electronic resonances of liquids are located at much higher energies than the photon energies used here. The almost concomitantly occurring second interaction with the pump field causes a $P_2$-like perturbation that leads to excitation of both the Lib($P_2$) and Reo($P_2$) modes.

Figure 5a suggests that thermal population of the various states may have a significant impact on the TKE signal. Indeed, since the THz dielectric susceptibility and the OKE signal of liquids can significantly depend on temperature[23], equation (3) implies the TKE signal does also. Therefore, varying the sample temperature in addition to the pump frequency is likely to deliver important information on the nature of the driven modes.

In conclusion, we have conducted a systematic comparison of the transient optical birefringence in various nonpolar versus strongly polar liquids following optical versus THz excitation. Our experimental observations of increased/decreased birefringence are fully consistent with the notion that THz fields exert ultrafast torque on molecules due to their coupling to the permanent molecular dipoles. Our observation of resonant THz excitation suggests significant applications, for instance, efficient molecular alignment of polar solutes and solvents, which may even enable coherent control of chemical reactions[37,38]. The THz field strength of $2\,\mathrm{MV\,cm^{-1}}$ used in this work can still be treated as a second-order perturbation of the liquid and results in a molecular alignment that is characterized by an averaged $\cos^2\theta - 1/3$ on the order of $10^{-4}$ (see 'Methods' section). However, massive alignment in the per cent range may come into reach with recently reported THz field amplitudes[39] of $> 10\,\mathrm{MV\,cm^{-1}}$.

From a spectroscopic viewpoint, resonant and selective excitation of THz motions can straightforwardly be extended to pairs of pump pulses[12,40–42]. In addition, temperature variation will allow one to systematically change the initial population of the various modes. Applied to hydrogen-bonded liquids like water, such temperature-dependent two-dimensional nonlinear

THz spectroscopy of transient optical birefringence will provide fundamental insights into the coupling of modes associated with the collective rotational and translational motion of hydrogen-bond networks. Finally, more sophisticated models will help reveal the role of interaction-induced fluctuating dipoles in the solvent dynamics.

## Methods

**Pump-probe set-up.** For the TKE measurements, intense THz fields at $\sim 1$ THz are generated by optical rectification of laser pulses (centre wavelength 800 nm, pulse duration 350 fs, pulse energy 4 mJ, repetition rate 1 kHz) from an amplified laser system in a 1.3 mol% MgO-doped stoichiometric LiNbO$_3$ crystal (LN) with the tilted-pulse-front technique[43,44]. To generate intense THz pulses at $\sim 2$ THz, pulses (1300 nm, 60 fs, 1 mJ) from an optical parametric amplifier are optically rectified in the organic crystal 4-N,N-dimethylamino-4′-N′-methyl-stilbazolium tosylate (DAST)[45].

In the experiment, the linearly polarized THz pump pulse is focused onto the sample cell (Fig. 1a). The induced transient birefringence is measured by a temporally delayed and collinearly propagating probe pulse whose incident linear polarization is set to an angle of 45° relative to the THz electric field (see Fig. 1a). Due to the pump-induced birefringence, the probe field components polarized parallelly ($\parallel$) and perpendicular ($\perp$) to the pump field acquire a phase difference $\Delta\phi$ when propagating through the sample, thereby resulting in elliptical polarization. The $\Delta\phi$ is detected with a combination of a quarter-wave plate and a Wollaston prism, which splits the incoming beam in two perpendicularly polarized beams with power $P_\parallel$ and $P_\perp$. In the limit $|\Delta\phi| \ll 1$, the normalized difference $P_\parallel - P_\perp$ fulfills

$$\frac{P_\parallel - P_\perp}{P_\parallel + P_\perp} = \Delta\phi \qquad (4)$$

and is measured by two photodiodes as a function of the temporal delay between THz pump and probe pulse[9,14]. For improved signal-to-noise ratio, the probe (2 nJ, 800 nm, 8 fs, 80 MHz) is derived from the low-noise seed laser oscillator rather than from the amplified output of the laser system[14].

The OKE experiments are performed in the same set-up as the TKE experiment, however, with the THz pump pulses replaced by optical pump pulses (800 nm, 350 fs, $\sim 2\,\mu$J) with identical linear polarization.

**Sample details.** The sample liquids are kept in static and flow cells. Here, the choice of the window materials is critical. Windows should be transparent at both THz (pump) and optical (probe) frequencies, optically isotropic and their nonlinear THz response should be small and short-lived[14]. To fulfil all these criteria, we employ 200 nm thick SiN membranes as windows for a static cell having a thickness of 100 $\mu$m.

To make sure that accumulation of pump heat does not influence the results, we performed the TKE experiments also in a flow cell with the same SiN windows. We found no difference between static and flow cells in terms of both dynamics and amplitudes of the signals, consistent with the less than 0.1 K temperature increase estimated for a single THz pump pulse.

The dried liquids were provided from Sigma-Aldrich and used as received. To avoid wetting of the liquids, preparation of liquids and experiments were done under N$_2$ purging. Stationary THz absorption spectra of liquids were obtained with a broadband THz time-domain spectrometer based on a broadband spintronic THz emitter[46].

**Signal normalization.** Our comparative method is based on the fact that the electronic response of liquids is identical at THz and optical pump frequencies because the associated photon energies ($\sim 1.5$ eV and $\sim 10$ meV, respectively) are much smaller than the electronic excitation energies ($> 5$ eV) of the liquids studied here.

By using a generic phenomenological model for the transient birefringence signal (Kerr effect), we show that nonpolar liquids (such as cyclohexane) exhibit identical TKE and OKE response (see Supplementary Note 1). Therefore, since both our optical and THz pump pulse have approximately identical shape (see inset of Fig. 2a), almost identical normalized birefringence dynamics result following optical and THz excitation of nonpolar liquids (see Fig. 2a and Supplementary Figs 2 and 3).

**Model details.** To develop a simple model that qualitatively describes the response of an ensemble of static rotors to an external electric field (optical and THz), we consider the dynamics of the angular distribution function $f(u,t)$. Here, $f(u,t)\mathrm{d}u$ quantifies the number of molecules having $u = \cos\theta$ in the interval $[u, u + \mathrm{d}u]$ at time $t$ where $\theta$ is the angle between the molecular dipole and the direction of the applied electric field (see Fig. 3). In equilibrium, $f$ equals $f_0 = N/2$, proportional to the particle density $N$ yet independent of $\theta$ (Fig. 3a).

In the rotational diffusion model, the equation of motion of $f(u,t)$ is given by[25]

$$\hat{O}f = C\partial_u\left[(T^{\mu_0} + T^{\Delta\alpha})f\right], \qquad (5)$$

where the $u$- and $t$-dependent operator $\hat{O}$ captures the dynamics of the system in the absence of external perturbations. The right-hand side of equation (1) describes the action of the external linearly polarized pump field having amplitude $\mathbf{E}(t)$ through the torques $T^{\mu_0}$ (mediated by the permanent dipole moment $\boldsymbol{\mu}_0$) and $T^{\Delta\alpha}$ (mediated by the field-induced electronic dipole moment $\boldsymbol{\mu}_{\mathrm{ind}}$). Consistent with equation (1), the torques are determined by[25]

$$T^{\mu_0} \propto \sin^2\theta\,\mu_0 E \quad \text{and} \quad T^{\Delta\alpha} \propto \Delta\alpha E^2 \sin\theta\cos\theta. \qquad (6)$$

Note that in the rotational diffusion model, $\hat{O}$ equals the sum of the Laplace operator and $(1/D)\partial_t$, the time-derivative normalized by the rotational diffusion constant $D$. The relevant eigenfunctions of the Laplace operator are given by the Legendre polynomials

$$P_0(u) = 1,\ P_1(u) = u,\ P_2(u) = (3u^2 - 1)/2,\ \dots \qquad (7)$$

It can be shown[25] that a single interaction with the pump field transfers population from the $P_0$-like ground state to $P_1$, whereas two interactions with the pump field will also populate $P_2$. This notion is consistent with the qualitative yet somewhat more general considerations underlying Fig. 3.

Because the rotational diffusion model is only capable of describing a random-walk-like relaxation of $f$ back to the equilibrium distribution $f_0$, we assume that additional rotational motions, such as ballistic rotation and librations, can be taken into account by an appropriately modified operator $\hat{O}$ (see ref. 47). Since we are primarily interested in comparing the ways the torques $T^{\mu_0}$ and $T^{\Delta\alpha}$ act on the system, we need not specify $\hat{O}$ further. The constant $C$ in equation (5) is eventually fixed by comparing the final result to the known solution for a static electric field[48].

Since in our experiment, the transient optical birefringence (equation (2)) has been found to scale quadratically with the incident pump field at both optical and THz excitation, we need to solve equation (5) up to second order in the applied electric field. Thus, the general solution has the structure

$$f = f_0 + \Delta f_1 + \Delta f_2, \qquad (8)$$

where $\Delta f_1$ and $\Delta f_2$, respectively, are contributions linear and quadratic in $E(t)$. By virtue of equation (5), $\Delta f_1$ and $\Delta f_2$ are found to obey the following differential equations[25],

$$\hat{O}\Delta f_1 = C\partial_u(T^{\mu_0}f_0), \qquad (9)$$

$$\hat{O}\Delta f_2 = C\partial_u\left(T^{\mu_0}\Delta f_1 + T^{\Delta\alpha}f_0\right). \qquad (10)$$

As $\Delta f_1$ and $\Delta f_2$, respectively, result from one and two interactions with the field of the pump pulse, it is often instructive to consider $\Delta f_1$ as the response to one $\delta$-like perturbation (Fig. 3a) and $\Delta f_2$ as the response to two subsequent $\delta$-like perturbations (Fig. 3b). From these impulse responses, the general linear and quadratic response can easily be determined.

As seen from the right-hand side of equation (9), $\Delta f_1$ arises from the perturbation $C\partial_u(T^{\mu_0}f_0)$, which is proportional to the first-order Legendre polynomial $P_1(u) = u = \cos\theta$. This $P_1$-type perturbation suggests the resulting response $\Delta f_1$ has also approximately $P_1$-like characteristics, $\Delta f_1(u,t) \propto P_1(u)$. Indeed, this assumption has been shown to be precisely valid for rotational diffusion[25,48], and it can be further bolstered by the schematic of Fig. 3a: in case of a $\delta$-like field pulse, a $\delta$-like torque is exerted which instantaneously increases the mean angular velocity of a molecule to a value proportional to the time-integrated torque, that is, $\sin\theta$. Therefore, as seen in Fig. 3a, population is shifted from the south pole ($\theta = 180°$) to the north pole ($\theta = 0°$), whereas it remains constant at the equator ($\theta = 90°$), consistent with a $\cos\theta$-type distribution change.

Note that the field-induced orientation of the molecular dipoles implies a polarization $P(t) \propto \mu_0 \int \mathrm{d}u\, P_1(u)\Delta f_1(u,t)$ along the field direction. This polarization is usually expressed by the convolution $P = \chi^{\mu_0} E$ where $\chi^{\mu_0}$ is the contribution of the permanent electric dipole moment $\boldsymbol{\mu}_0$ to the familiar linear dielectric susceptibility $\chi$. Note that $\chi$ can be measured by microwave and THz absorption spectroscopy[23] but also contains contributions from other (for example, translational) degrees of freedom. The $\chi^{\mu_0}(t)$ captures the full dynamics of the $P_1$-component of $\Delta f_1$, from the build-up (for example, due to initial ballistic rotation) to the final decay (for example, due to rotational diffusion). Thus, the change in the angular distribution function arising from a first interaction with the field of the pump pulse can be approximated as

$$\Delta f_1(u,t) \propto P_1(u) \cdot (\chi^{\mu_0} E)(t). \qquad (11)$$

According to equation (10), the second-order response arises from the term $C\partial_u(T^{\mu_0}\Delta f_1 + T^{\Delta\alpha}f_0)$, which is proportional to the second-order Legendre polynomial $P_2(u) = (3u^2 - 1)/2$. Therefore and analogous to the linear case, we assume this perturbation leads to a $P_2$-like change in the distribution function, both for the $\boldsymbol{\mu}_0 \times \mathbf{E}$- (Fig. 3b) and $\boldsymbol{\mu}_{\mathrm{ind}} \times \mathbf{E}$-type torque (Fig. 3c). The temporal dynamics of $\Delta f_2$ are described by the response function $R_2$, which captures the build-up and (possibly oscillatory) decay of an impulsively induced $P_2$ distribution.

The $R_2(t)$ can, in principle, be measured by the ultrafast OKE because optical fields induce exclusively and impulsively the $P_2$-like perturbation $C \, \partial_u (T^{\Delta\alpha} f_0)$.

By evaluating the right-hand side of equation (10) by means of equation (11), subsequent convolution with $R_2$ yields

$$\Delta f_2(u,t) \propto P_2(u) \cdot \{R_2 \left[ E \cdot N \Delta\alpha E + 3E \cdot (\chi^{\mu_0} E) \right]\}(t). \qquad (12)$$

This result is consistent with the special case of rotational diffusion, which delivers mono-exponentially decaying step functions $\chi^{\mu_0}(t) = 2D\chi^{\mu_0}_{DC}\Theta(t)\exp(-2Dt)$ and $R_2(t) \propto \Theta(t)\exp(-6Dt)$ for the two response functions[25]. Here, $D$ is the diffusion constant, $\Theta(t)$ is the Heaviside step function and $\chi^{\mu_0}_{DC} = N\mu_0^2/3\varepsilon_0 k_B T$ is the susceptibility for a static electric field with $\varepsilon_0$ and $k_B T$ being the vacuum permittivity and thermal energy, respectively. Finally, by projecting $\Delta f_2(u,t)$ (equation (12)) onto $P_2(u)$ (see equation (2)), we obtain the optical birefringence $\Delta n(t)$ (see equation (3)).

**Alignment estimate.** In the case of negligible velocity mismatch between the pump and probe pulses, the birefringence-induced phase shift experienced by the probe is given by $\Delta\phi = 2\pi L\Delta n/\lambda$ where $L = 100\,\mu m$ is the sample thickness and $\lambda = 800\,nm$ is the probe centre wavelength[9]. Therefore, in DMSO (see Fig. 2b), the peak $\Delta n$ amounts to $\sim 8 \times 10^{-6}$.

On the other hand, the refractive index anisotropy $\Delta n = n_\parallel - n_\perp$ is connected to the mean $P_2(u)$ by $2n\Delta n = n_\parallel^2 - n_\perp^2 = \Delta\alpha N \langle P_2(u) \rangle$ (see refs 6,13 and equation (2)). With the refractive index $n = 1.5$, the molecule density $N = 8 \times 10^{27}\,m^{-3}$ (ref. 49) and polarizability anisotropy $\Delta\alpha = 4\pi \times 1.7 \times 10^{-30}\,m^3$ (refs 20,49), we obtain $P_2(u) = 1.3 \times 10^{-4}$ at a THz field amplitude of $2\,MV\,cm^{-1}$ (see Fig. 2b).

**Data availability.** The data sets generated and analysed during the current study are available from the corresponding authors on reasonable request.

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

## Acknowledgements

We thank R.K. Campen and N.P. Ernsting for stimulating discussions. T.K. acknowledges the European Research Council for support through Grant No. 681917

(TERAMAG) and the German Research Foundation for funding through Grant No. KA 3305/2-1.

## Author contributions

M.S. and T.K. conceived the experiments. M.S. built the set-up, performed the experiments and analysed the experimental data with contributions from T.K. and M.W. T.K. and M.S. developed the analytical model. All the authors contributed to discussing the results and writing the paper.

## Additional information

**Competing interests:** The authors declare no competing financial interests.

**Publisher's note**: 

DOI: 10.1038/ncomms15796    OPEN

# Erratum: Transient birefringence of liquids induced by terahertz electric-field torque on permanent molecular dipoles

Mohsen Sajadi, Martin Wolf & Tobias Kampfrath

*Nature Communications* 8:14963 doi: 10.1038/ncomms14963 (2017); Published 10 Apr 2017; Updated 25 May 2017

This Article contains typographical errors in six instances in which the convolution symbol '∗' was mistakenly omitted from six equations during the production stage. Each instance is described below.

The penultimate sentence of the first paragraph of the Results subsection titled 'Model' should read:

'Therefore, $\Delta f_1$ does not yet cause optical birefringence but is accompanied by a time-dependent dielectric polarization $\mathbf{P} = \chi^{\mu_0} * \mathbf{E}$.'

The correct form of equation (3) is:

$$\Delta n(t) \propto R_2 * [E \cdot (N\Delta\alpha E + 3\chi^{\mu_0} * E)]. \tag{3}$$

In the second paragraph after equation (3), the first sentence should read:

'Note that equation (3) reveals an analogy of the $\mathbf{\mu}_{ind}$- and $\mathbf{\mu}_0$-related coupling mechanisms: the first field interaction generates an effective electronic ($N\Delta\alpha\mathbf{E}$) and orientational polarization ($\chi^{\mu_0} * E$) which, in turn, serves as a handle for the second field interaction to generate a $P_2$-like perturbation (square bracket in equation (3)).'

In the paragraph preceding equation (11), the second sentence should read:

'This polarization is usually expressed by the convolution $P = \chi^{\mu_0} * E$ where $\chi^{\mu_0}$ is the contribution of the permanent electric dipole moment $\mathbf{\mu}_0$ to the familiar total dielectric susceptibility $\chi$.'

The correct form of equation (11) is:

$$\Delta f_1(u, t) \propto P_1(u) \cdot (\chi^{\mu_0} * E)(t). \tag{11}$$

The correct form of equation (12) is:

$$\Delta f_2(u, t) \propto P_2(u) \cdot \{R_2 * [E \cdot N\Delta\alpha E + 3E \cdot (\chi^{\mu_0} * E)]\}(t) \tag{12}$$

