## [Peer Review File · Nature Communications]

Reviewers' comments:

Reviewer #1 (Remarks to the Author):

The authors present a combination of Optical and THz Kerr Effect (OKE and TKE) measurements of several liquids. The results show a difference in the OKE vs. TKE data for polar liquids, whereas the measured data are the same for non-polar (or, more precisely here for toluene and cyclohexane, weakly polar) liquids. The authors conclude that these results show that the THz field can couple to the permanent dipole moment of the molecule and drive molecular orientation. While the spectroscopic data herein are of very high quality and nicely modeled, the paper suffers from several flaws in interpretation and a systematic undervaluing of previously published literature. The comparison of OKE and TKE techniques in particular is a worthwhile study, and clearly meets the bar for publication in more specialized journals. However, as presently written, this work provides little in the way of new physical (or chemical) insight, and given the issues outlined below, I find the manuscript to be well below the expected level of general impact and broader interest expected of publications in Nature Communications. After considerable corrections, the manuscript would be highly appropriate for a field-specific journal, such as J. Chem. Phys.

Major Issues:

1. The claim that THz light can drive molecular orientation is well established in the literature. Beyond the Hoffman et al. 2009 paper cited, which is also the subject of a Nature News & Views, <http://www.nature.com/nphoton/journal/v4/n3/full/nphoton.2010.14.html>, there are many additional results from the Nelson group at MIT: S. Fleischer, Y. Zhou, R.W. Field, and K.A. Nelson, Phys. Rev. Lett. 107, 163603 (2011). S. Fleischer, R.W. Field and K.A. Nelson, Phys. Rev. Lett. 109, 123603 (2012). Extensions of these approaches in liquids are found in the Finneran et al. PNAS reference cited, but also in Allodi et al. J. Chem. Phys. 143, 234204 (2015). This latter work presents data on THF, and an electronic + orientational/ alignment model in addition to the vibrational coherences that are main focus of this manuscript. And, the Finneran et al. work demonstrates control over the molecular alignment using two fs-THz pulses. Thus, major sections of the paper present results in a fashion that do little justice to the work reported previously. They are not a sufficiently novel results, on their own, to warrant publication.

2. The comparison of polar species with different polarizabilities and with non-polar species (esp. Figure 2) is a nice step forward, and brings welcome order to points that were only hinted at in Hoffman et al. A focus here, with a proper overview of the TKE work done to date, would greatly improve the manuscript.

SOMEWHAT LESS CRITICAL COMMENTS:

3. THz pulses are too broad in frequency to make the carrier envelop approximation, reliably, and so characterizing the pulses as centered at 1 vs. 2 THz may have little physical meaning. The spectral profiles are indeed different but both have significant frequency content more than an octave away from the noted frequency. This will surely affect the results, especially since the LiNbO3 pulses have a smaller bandwidth than those produced by DAST. Numerical simulations of the molecular dynamics, including the appropriate pulse shapes, would be far more realistic.

3. The paper claims that THz and optical excitation drive the same modes of the liquid. While in some sense this is true, in a larger sense, it obscures the point that THz light can act via a direct dipolar mechanism whereas optical light must go through a Raman process. Thus the selection rules are different, and statements such as this muddle the larger point of the paper.

MINOR COMMENTS:

4. In a technical vein, are the 800 nm pulses stretched here to optimize the conversion efficiency of the tilted pulse front scheme? Given the superb Ti:Sapph oscillator temporal pulse duration, the statement in the Methods 'since both our optical and THz pump pulse have approximately identical shape', it would appear so.

5. I would encourage the authors to go through the manuscript carefully for typographical errors and those of grammatical usage, of which there are many.

Reviewer #2 (Remarks to the Author):

This is a very nice manuscript describing transient birefringence measurements in the optical and terahertz domains, highlighting the similarities and differences which arise in association with the presence (or lack) of a permanent dipole moment. I find this work to be original and compelling, and feel that it will be of significant interest to more than one research community. Therefore I strongly recommend publication.

A few minor questions for the authors are in order:

1. The manuscript implies that the optical and terahertz Kerr effect dynamics should be identical in the case of a non-polar liquid. However, I would expect a small difference. These non-polar liquids are not perfectly transparent in the terahertz range - there is a small absorption coefficient, usually attributed to transient collision-induced dipoles. I would expect these transient dipoles to have some (perhaps small) influence on the terahertz Kerr signal, which would not be present in the optical signal. Perhaps the authors could comment on this possibility?
2. The data shown in Fig. 4 are a bit puzzling. The authors point out that the observed trend is opposite to that which might naively be anticipated, and attribute this to the excitation of librational modes in DMSO. This supposition would be more strongly supported by seeing similar data from other solvents, e.g., one where the librations lie at much higher frequency. Would the authors anticipate a different result in that case?

Reviewer #3 (Remarks to the Author):

The manuscript by Sajadi, Wolf and Kampfrath reports a very interesting experimental study comparing the ultrafast transient birefringence upon optical and THz excitation at the example of a range of organic liquids. Depending on the type of aprotic polar liquid the dynamics of the birefringence vary systematically at THz frequencies compared to the optical excitation. Together with the experimental data the authors present a model to qualitatively explain the observed effects. Overall the paper presents novel insights based on a highly original experimental and sound theoretical analysis.

The manuscript is well written, concise and very clear. The experimental data are of excellent quality, the overall discussion of the work is sound and the results and conclusions are convincing.

The authors suggest that the data presented indicates librational and reorientational motions in DMSO are coupled. This is certainly a strong possibility and is supported by previous work in other groups. It would be useful if the authors could expand their discussion a bit further and take into account also the effects of secondary dielectric relaxations that have been suggested to play a role at THz frequencies for liquids. In this context it would also be interesting to see the role of the molecular internal degrees of freedom on the response. The model systems chosen are very simple and the theoretical model matches this simplicity - this is not a criticism in the work presented, as it is very helpful indeed to prove the effect using a simple system, rather I wonder whether the authors might want to comment on this aspect. Have you studied larger molecules that exhibit greater intramolecular flexibility? I would expect even stronger differences between the THz and optical response and it would be interesting to see such results.

The paper briefly discusses the topic of radiation induced net heating effects in the experimental section. Have you investigated the role of transient heating at picosecond timescales on the measured birefringence dynamics? Clearly the magnitude of absorption of the EM wave is different

at the two frequency ranges studied and it would be very useful to see a discussion of this effect.

Please add the experimental details of the OKE setup used. The manuscript currently only describes the THz measurements.

Caption Figure 2: remove 'to the birefringence' in line 3.

Reviewer #1 (Remarks to the Author):

The authors present a combination of Optical and THz Kerr Effect (OKE and TKE) measurements of several liquids. The results show a difference in the OKE vs. TKE data for polar liquids, whereas the measured data are the same for non-polar (or, more precisely here for toluene and cyclohexane, weakly polar) liquids. The authors conclude that these results show that the THz field can couple to the permanent dipole moment of the molecule and drive molecular orientation. While the spectroscopic data herein are of very high quality and nicely modeled, the paper suffers from several flaws in interpretation and a systematic undervaluing of previously published literature.

The comparison of OKE and TKE techniques in particular is a worthwhile study, and clearly meets the bar for publication in more specialized journals. However, as presently written, this work provides little in the way of new physical (or chemical) insight, and given the issues outlined below, I find the manuscript to be well below the expected level of general impact and broader interest expected of publications in Nature Communications. After considerable corrections, the manuscript would be highly appropriate for a field-specific journal, such as J. Chem. Phys.

Response: We would like to thank the reviewer for her/his interest in our work and her/his detailed and useful comments that have led to a significantly improved manuscript. As detailed point by point in the following, we also elaborate on the novelty and added value of our work, in particular with respect to previous works.

Major Issues:

1. The claim that THz light can drive molecular orientation is well established in the literature. Beyond the Hoffman et al. 2009 paper cited, which is also the subject of a Nature News & Views, <http://www.nature.com/nphoton/journal/v4/n3/full/nphoton.2010.14.html>, there are many additional results from the Nelson group at MIT: S. Fleischer, Y. Zhou, R.W. Field, and K.A. Nelson, Phys. Rev. Lett. 107, 163603 (2011). S. Fleischer, R.W. Field and K.A. Nelson, Phys. Rev. Lett. 109, 123603 (2012). Extensions of these approaches in liquids are found in the Finneran et al. PNAS reference cited, but also in Allodi et al. J. Chem. Phys. 143, 234204 (2015). This latter work presents data on THF, and an electronic + orientational/ alignment model in addition to the vibrational coherences that are main focus of this manuscript. And, the Finneran et al. work demonstrates control over the molecular alignment using two fs-THz pulses. Thus, major sections of the paper present results in a fashion that do little justice to the work reported previously. They are not a sufficiently novel results, on their own, to warrant publication.

Response: We agree with the reviewer that we need to more clearly differentiate the results of our work from previous THz works. We also regret that our previous manuscript apparently created the impression of “systematic undervaluing of previous work”. This was certainly not our intent. Actually, we had cited most of the works (3 out of 4) mentioned by the reviewer, but did not discuss them in detail. This has been changed in the new manuscript.

We have split our response to this referee’s point in three subsections: our key results, their relevance and their comparison to previous works.

Our key results. Our work provides the following key results:

- 1) We observe transient birefringence in polar liquids induced by torque on solvent molecules due to THz-field coupling to (i) the permanent molecular dipole moment μ_0 and/or (ii) the induced electronic dipole moment μ_{ind} .
- 2) We separate these contributions by a new experimental strategy: we compare THz and optical excitation of the liquid. In the case of optical excitation, only μ_{ind} is operative because the μ_0 -torque is too fast for the molecule to follow.
- 3) Frequency-dependent measurements indicate that the enhanced coupling due to μ_0 -torque can be understood as a resonant Raman process with a librational mode as an intermediate state (please see the new Figs. 4c,d). Thus, we find indications of coupling between librational and reorientational modes of the solvent molecules.

- 4) We develop a simple model, an extended version of the rotational diffusion model. This model provides a consistent qualitative description of our experimental results, even when reorientational motion beyond rotational diffusion (for example librations) is considered.

All points are new and, in our opinion, highly relevant for a general understanding of collective molecular dynamics in liquids as detailed in the next paragraph.

Relevance of our results. First, the results 1) and 2) are an important step toward disentangling coupling mechanisms of liquids and strong THz electric fields.

Second, the results 1) and 2) demonstrate a new important handle to molecular motion. So far, the permanent dipole moment μ_0 of solvent molecules was only accessible by linear dielectric and/or THz spectroscopy. Our work shows that μ_0 can also be addressed in a nonlinear manner with respect to the THz field E (the optical birefringence scales with E^2). In this way, μ_0 becomes accessible by nonlinear THz spectroscopy, which is expected to provide considerable insights into the coupling of intermolecular modes of liquids. A first important glimpse of this potential is provided by result 3) which hints to the coupling of librational and reorientational modes.

Third, for liquids with small optical Raman cross sections (such as DMSO), the OKE signal from low-frequency modes is typically very weak. Our results show that for polar liquids, THz excitation drives a resonant Raman process which is attractive for applications that require stronger Kerr signals.

Fourth, our modeling will help guide future THz-induced optical birefringence experiments, in particular in the form of two-dimensional spectroscopy. For example, our new Fig. 4b (TKE vs OKE data of acetonitrile) provides further support for our interpretation that a librational mode is primarily excited and then converted into reorientational motion. Better spectral resolution (e.g. by 2D THz spectroscopy) will shed more light on this important point.

Finally, our results can be regarded as the first step towards molecular alignment in liquids, with the ultimate goal of coherent control over chemical reactions using intense THz pulses.

Therefore, we strongly believe that our results are definitely new and interesting for several communities (e.g. condensed-matter physics and nonlinear THz spectroscopy), for which Nature Communication is the ideal outlet.

Complementarity with respect to previous works. We fully agree with the reviewer that there are previous works that deal with THz-field-induced optical birefringence in gases and liquids. In the following, we detail why our key results are complementary and new and have only little overlap with previous works.

1) Nelson and colleagues [M. Hoffmann, et al., APL 95, 231105 (2009)] applied THz pulses to liquids and observed instantaneous and longer-lived birefringence signals, the latter of which were assigned to reorientational motions of molecules. However, this paper focuses on nonpolar liquids. Regarding polar liquids, the authors write on p.231105-3: "So far we have not observed measurable optical birefringence in polar liquids [...]". Therefore, this paper does not consider coupling of THz electric fields to permanent electric molecular dipoles, which, in contrast, is the central topic of our work (please see points 1)-3) above).

2) Nelson and colleagues [S. Fleischer et al., PRL 107, 163603 (2011) & S. Fleischer et al., PRL 109, 123603 (2012)] focus on molecules in the gas phase and on coherent effects, in particular the revival of rotational quantum-mechanical wave packets. In contrast, our work deals with liquids, which exhibit considerably stronger intermolecular couplings and, thus, often show more complex molecular motions than gas phase molecules.

3) Blake and colleagues [Allodi et al. J. Chem. Phys. 143, 234204 (2015)] focus on oscillatory, intramolecular modes excited by THz radiation. The authors also observe longer-lived signal components which are discussed briefly and assigned to reorientational motions. A more detailed discussion of this feature is postponed ("The use of the biexponential model and the intermediate response will be discussed in more detail in an upcoming publication."). In contrast, such signatures of reorientational motions are in the focus of our work, in particular its excitation by coupling to the permanent and induced electronic dipole moment of the molecule.

4) Blake and colleagues [A. Finneran et al., PNAS 113 6857, (2016)]: this pioneering work demonstrates THz-THz-2D-Raman spectroscopy by using two time-delayed THz pump pulses. This study demonstrates the feasibility of TKE to identify the anharmonic coupling of thermally populated *intramolecular* vibrational modes. In addition, a longer-lived, non-oscillatory feature is observed and assigned to reorientational motion. To characterize the transient signal by only a few parameters, a phenomenological formula (a convolution of the two THz pump fields with an instantaneous plus an exponentially decaying response kernel) is used. These results are complementary to our results 1)-3) above because we in particular study the coupling of THz waves to the permanent and induced electronic dipole moment of the molecule. It is extremely interesting to transfer this 2D technique to our approach.

Action: We modified the manuscript as follows: we better explain our key results and their relevance according to our response above (please see the abstract, the introduction and conclusion).

We added a section to the introductory section in which previous nonlinear THz studies of liquids are discussed, finally leading to the focus of our work.

2. The comparison of polar species with different polarizabilities and with non-polar species (esp. Figure 2) is a nice step forward, and brings welcome order to points that were only hinted at in Hoffman et al. A focus here, with a proper overview of the TKE work done to date, would greatly improve the manuscript.

Response: We fully agree with the reviewer that the focus of the current manuscript is on the coupling of THz electric fields to the permanent dipole moment of polar molecules in liquids. As discussed above, this focus needs to be better differentiated from previous THz pump/optical probe works on liquids.

Action: In the abstract, the introduction and the conclusion section, we now better explain the focus of our work, our key results, their novelty and their relevance.

We added text to the introduction in which previous THz works are discussed.

SOMEWHAT LESS CRITICAL COMMENTS:

3. THz pulses are too broad in frequency to make the carrier envelop approximation, reliably, and so characterizing the pulses as centered at 1 vs. 2 THz may have little physical meaning. The spectral profiles are indeed different but both have significant frequency content more than an octave away from the noted frequency. This will surely affect the results, especially since the LiNbO3 pulses have a smaller bandwidth than those produced by DAST. Numerical simulations of the molecular dynamics, including the appropriate pulse shapes, would be far more realistic.

Response: The reviewer is right that the center frequency of broadband pulses may not be sufficient for their characterization. This point is particularly relevant for Supplementary Fig. S5, in which the TKE vs OKE signal enhancement is plotted as a function of frequency, i.e. for THz pump pulses of relatively small bandwidth.

Thus, for the very broadband pulses in our experiments, we have calculated the transient birefringence by using Eq. (3) (main text) and the transient electric field of the THz pump pulses, both from the DAST and LN sources (see Fig. S6). The result is shown in Fig. S5b. As expected from the trend of Fig. S5a, increased birefringence amplitude is obtained for the LN pump, whose spectrum is centered at lower frequencies (~1 THz) than the DAST spectrum (~2 THz, see Fig. S5a).

Action: We added text (section 4) and a new figure (Fig. S5b) to the Supplementary Information.

3. The paper claims that THz and optical excitation drive the same modes of the liquid. While in some sense this is true, in a larger sense, it obscures the point that THz light can act via a direct dipolar mechanism whereas optical light must go through a Raman process. Thus the selection rules are different, and statements such as this muddle the larger point of the paper.

Response: 1) We agree with the reviewer that the statement “THz and optical pump drive the same modes of the liquid” could be perceived as overly generalized. What we meant to say was that the only modes (i.e. the relaxation tail of the transient birefringence signal) we observe in our experiment are

found to be excited by both optical and THz pumping. This of course leaves open the possibility that other modes are exclusively driven by either optical or THz excitation.

2) We are grateful to the reviewer for bringing up the important points of Raman processes and selection rules: driving modes by a one-photon infrared (IR) and a two-photon Raman mechanism generally obeys different selection rules. While direct IR excitation relies on one interaction with the pump field, a Raman process relies on two interactions, which is, for example, necessary to rectify the too rapidly oscillating light field.

In our experiments, we observe that both optical and THz pumping excite the same dynamics (modes). In both cases, the mode amplitude scales with the squared pump field E^2 . This shows that the observed mode is formally Raman-active and driven by a Raman-type process for *both* optical and THz pumping.

The surprising point of our observations is that in polar liquids and by using THz pump pulses, the excitation process is enhanced. The results show that the enhancement is related to the torque the THz pulse exerts through coupling to the permanent molecular dipole moments. Our measurements on DMSO (Fig. 4a) using pump pulses with different center frequency indicate that the enhancement arises from resonant interaction with the librational mode. This notion of a resonant intermediate state is supported by new measurements on another polar liquid, acetonitrile, which has a prominent IR-active librational mode at about 3 THz (please see new Fig. 4b). This process can be considered as a resonant process with a librational mode as intermediate state (see schematic of new Fig. 5a).

Triggered by a comment from reviewer#3, there is a second excitation/relaxation pathway that can in principle contribute to the observed enhancement of the reorientational mode amplitude. Through mode coupling with the reorientational mode, the librational mode can partly be transformed into reorientational motion. Please see the schematic of the new Fig. 5a and our response to reviewer#3 below.

The schematic seen in the new Figs. 5a,b summarizes our interpretation: in our experiment, THz and optical pump pulses are observed to eventually excite the same reorientational mode $Reo(P_2)$ by means of two optical transitions (see new Fig. 5a). For optical pumping, there is no resonant intermediate state because the light field couples primarily to the electrons which assume an instantaneous induced electronic dipole moment μ_{ind} . For THz excitation, in contrast, not only this μ_{ind} -pathway is operative. In addition, coupling to the permanent molecular dipole moment μ_0 can excite IR-active intermediate librational modes $Lib(P_1)$, which can enhance the amplitude of the finally measured mode $Reo(P_2)$.

This interpretation is also reflected by Eq(3), the central result of our modeling: transient birefringence results from coupling through the induced electronic dipole moment μ_{ind} (scaling with polarizability anisotropy $\Delta\alpha$) and/or the permanent dipole moment μ_0 . More generally, as discussed in e.g. [Boyd, Nonlinear Optics], both the dielectric response ($\chi^{(1)}$) and TKE ($\chi^{(3)}$) can be enhanced by the same optical transition into a resonant final/intermediate state.

Action:

1) In section “Transient birefringence”, we more precisely and restrictively say that all modes observed in our experiment are found to be excited by both optical and THz pumping.

2) We added new figure panels to bolster our interpretation with additional data (Fig.4b) and to illustrate it by means of schematics (Fig.5). We added text (section “Discussion”) similar to the discussion above.

MINOR COMMENTS:

4. In a technical vein, are the 800 nm pulses stretched here to optimize the conversion efficiency of the tilted pulse front scheme? Given the superb Ti:Sapph oscillator temporal pulse duration, the statement in the Methods 'since both our optical and THz pump pulse have approximately identical shape', it would appear so.

Response: The 800nm pulses were stretched on purpose to achieve an instantaneous intensity that has an approximately identical shape as the instantaneous intensity $E^2(t)$ of the THz pulse. In this way, comparison of normalized OKE and TKE data becomes straightforward.

Action: We added text (section “Experiment”) to better explain our motivation for using stretched pulses: “... thereby allowing for straightforward comparison of TKE and OKE data”.

5. I would encourage the authors to go through the manuscript carefully for typographical errors and those of grammatical usage, of which there are many.

Response: We have carefully checked all the text and figures. We have fixed typos, potentially ambiguous figure labeling and language. Several test readers (native English speakers) rate the new manuscript as very readable and clear.

Reviewer #2 (Remarks to the Author):

This is a very nice manuscript describing transient birefringence measurements in the optical and terahertz domains, highlighting the similarities and differences which arise in association with the presence (or lack) of a permanent dipole moment. I find this work to be original and compelling, and feel that it will be of significant interest to more than one research community. Therefore I strongly recommend publication.

Response: We would like to thank the reviewer for her/his interest in our work and her/his detailed, useful and encouraging comments that have led to a significantly improved manuscript. In the following, we address all of the reviewer's points in detail.

A few minor questions for the authors are in order:

1. The manuscript implies that the optical and terahertz Kerr effect dynamics should be identical in the case of a non-polar liquid. However, I would expect a small difference. These non-polar liquids are not perfectly transparent in the terahertz range - there is a small absorption coefficient, usually attributed to transient collision-induced dipoles. I would expect these transient dipoles to have some (perhaps small) influence on the terahertz Kerr signal, which would not be present in the optical signal. Perhaps the authors could comment on this possibility?

Response: We thank the reviewer for bringing up this very interesting point. If absorption of the THz pump by collision-induced transient electric dipoles did make a strong contribution to the pump-induced optical birefringence, we should observe a different normalized birefringence decay for THz and optical excitation (i.e. TKE vs OKE). However, within the uncertainties of our approach, we observe an identical normalized birefringence decay following both THz and optical excitation in three nonpolar liquids: toluene, cyclohexane and hexane (data for the latter not shown in the paper). We therefore conclude that the contribution of THz absorption by collision-induced transient electric dipoles makes a negligible contribution to the birefringence signal in our experiment.

It would be of course interesting to look for signatures of the excitation of such transient dipoles. To isolate their contribution to the TKE signal, we need to look for probably very small differences between deconvoluted OKE and TKE signals. This deconvolution challenge could be simplified by e.g. using very thin samples (such that pump-probe velocity mismatch becomes negligible), which, unfortunately, comes at the price of a considerably reduced signal strength. Accordingly, we are currently working on further improving the TKE/OKE signal-to-noise ratio.

To observe collision induced effects (i.e. the isotropic part of the $\chi^{(3)}$ response), it may be also helpful to make use of polarization-selective schemes (please see, e.g., Tokmakkof et al., *Chem. Phys. Lett.* **321**, 231 (2000). 231–237).

Action: We added text (section “Nonpolar vs polar liquids”) that ends with the conclusion that “that the weak THz-pump absorption by collision-induced transient electric dipoles makes a negligible signal contribution within the accuracy of our normalization procedure.”

2. The data shown in Fig. 4 are a bit puzzling. The authors point out that the observed trend is opposite to that which might naively be anticipated, and attribute this to the excitation of librational modes in DMSO. This supposition would be more strongly supported by seeing similar data from other solvents, e.g., one where the librations lie at much higher frequency. Would the authors anticipate a different result in that case?

Response: Following the suggestion by the reviewer, we conducted TKE/OKE measurements on another liquid. We chose acetonitrile because its dielectric spectrum up to 3 THz (which is the range accessible with our THz pump sources) features three separate contributions: a Debye-type resonance (around $1/(2\pi \cdot 3.3 \text{ ps}) = 50 \text{ GHz}$), translational motions (around 0.5 THz) and librations (around 3 THz). As with DMSO, we observe enhancement of the TKE vs OKE signals when we shift the pump center frequency from 0.7 to 2 THz, i.e. toward the librational mode. This enhancement is even more pronounced than in DMSO.

This additional measurement supports our suggestion that resonant excitation of the librational modes leads to enhanced birefringence. This interpretation is consistent with our modeling. As seen from

Eq(3), an enhanced susceptibility $\chi^{\mu 0}$ (which is the permanent-dipole-torque-related contribution to the total susceptibility) should result in enhanced amplitude of the resulting optical birefringence.

Resonant THz excitation of the libration resonantly enhances the P_2 -like-perturbation of the liquid and thus leads to an enhanced amplitude of reorientational motion. As schematically shown in the new Fig. 5a, this mechanism can be understood as a stimulated process with a resonant intermediate state (a librational mode), in contrast to optical excitation which has no resonant intermediate state.

It is very interesting to elucidate the role of the translational motions around 0.5 THz. This requires better frequency resolution with respect to the pump beam. Accordingly, 2D-type spectroscopy is currently being set up in our laboratory. Alternatively, as suggested by the reviewer, one can opt for polar liquids with librational modes located at much higher frequencies. In this case, librations and lower lying translational modes (so-called secondary dielectric relaxation modes) possibly have a better spectral separation.

Action:

We added TKE/OKE data on acetonitrile to Fig. 4 (now Fig. 4b, DMSO is shown by Fig. 4a) and added text to explain and discuss the new acetonitrile data (section "Discussion").

To illustrate the (off-)resonant transitions involved in optical and THz excitation as well as a subsequent relaxation process, we added schematics (see new Figs. 5a,b).

Reviewer #3 (Remarks to the Author):

The manuscript by Sajadi, Wolf and Kampfrath reports a very interesting experimental study comparing the ultrafast transient birefringence upon optical and THz excitation at the example of a range of organic liquids. Depending on the type of aprotic polar liquid the dynamics of the birefringence vary systematically at THz frequencies compared to the optical excitation. Together with the experimental data the authors present a model to qualitatively explain the observed effects. Overall the paper presents novel insights based on a highly original experimental and sound theoretical analysis.

The manuscript is well written, concise and very clear. The experimental data are of excellent quality, the overall discussion of the work is sound and the results and conclusions are convincing.

Response: We would like to thank the reviewer for her/his interest in our work and her/his detailed, useful and encouraging comments that have led to a significantly improved manuscript. In the following, we address all of the reviewer's points in detail.

The authors suggest that the data presented indicates librational and reorientational motions in DMSO are coupled. This is certainly a strong possibility and is supported by previous work in other groups. It would be useful if the authors could expand their discussion a bit further and take into account also the effects of secondary dielectric relaxations that have been suggested to play a role at THz frequencies for liquids.

Response:

1) We are grateful to the reviewer for bringing this aspect up because it pointed us to a scenario (suggested by Fayer and colleagues in *J. Chem. Phys.* 90, 6893-6902, (1989)) that describes a microscopic link between librational and reorientational motion. We now discuss this mode-coupling mechanism and another possible scenario we suggest for our experiment.

A) Fayer and colleagues [Deeg, F. W. et al., *J. Chem. Phys.* 90, 6893-6902, (1989)] excited liquids optically and so exerted an impulsive P_2 -type perturbation on the angular distribution function $f(u=\cos\theta, t)$ as schematically shown by the new Fig. 5b. The resulting birefringence dynamics was interpreted as a coupling of librational motion into reorientational diffusion. In this model, relaxation of the libration arises because of collisions between a molecule and its cage. Since the solvent cage changes in the course of time (known as secondary β relaxation), the librating molecule inside the cage does not relax to its initial direction. Thus, alignment remains which relaxes by diffusive reorientational motion. According to Deeg et al., the efficiency of this coupling process is the larger, the closer the time scales of librational damping and cage relaxation are. Please note that in this model, the coupling between libration and reorientation does not require the presence of a pump electric field (see new Fig. 5b).

B) Our model is complementary as it is concerned with the pump-induced excitation of modes and therefore requires the presence of a pump field. In this sense, it considers pump-field-induced coupling between librational and reorientational motion. More precisely, when the pump predominantly acts on the permanent molecular dipole, the first interaction with the pump leads to a P_1 -like perturbation $\Delta f_1(u, t)$ of the angular distribution function $f(u, t)$ (please see Fig. 3a). The resulting dynamics can be both reorientational and librational motion with P_1 character. In the second interaction with the pump field, the pump electric field and the nonzero $\Delta f_1(u, t)$ altogether lead to a P_2 -like perturbation of the distribution function which again are capable of triggering both reorientational and librational motion, but now with P_2 character (see e.g. the right-hand side of Eq.(9)). We note that this two-field excitation process can be interpreted as a resonant stimulated process (please see the new Figs.5a for a schematic).

While model B) is focused on the way the pump field exerts torques on molecular dipoles, the Fayer model A) deals with the coupling of librational motion into reorientational motion in the absence of

the pump field. Triggered by the reviewer's question we note that both scenarios can contribute to the reorientational relaxation tail of our TKE data, as schematically shown by the new Fig. 4d:

(i) Due to B), the E-field driven transition sequence

excites the P_2 -type reorientational motion $\text{Reo}(P_2)$ (see arrows 1 and 2a in Fig. 4d).

(ii) Alternatively, the sequence

excites the P_2 -type libration $\text{Lib}(P_2)$ which then drives reorientational motion $\text{Reo}(P_2)$ due to mechanism A) (see arrows 1 and 2b in Fig. 4d).

In both pathways (i) and (ii), the intermediate state $\text{Lib}(P_1)$ resonantly enhances the process. This is only possible with THz pump fields which couple resonantly to these modes through the permanent molecular dipole moment μ_0 (see arrow 1 in Fig. 4d).

The relative strength of pathways (i) vs (ii) is determined by the branching ratio of the transition from $\text{Lib}(P_1)$ to $\text{Reo}(P_2)$ vs $\text{Lib}(P_2)$. To measure this ratio, pump deconvolution is required (to also reveal the fast libration dynamics) as well as a 2D pump scheme (to frequency-resolve the THz pump process). We are currently extending our setup in this direction.

2) To further support the notion that resonant excitation of librational modes enhances transient optical birefringence and to address the, we conducted TKE/OKE measurements on another liquid, acetonitrile because its dielectric spectrum up to 3 THz (which is the range accessible with our THz pump sources) features three separate contributions: a Debye-type resonance (around $1/(2\pi \cdot 3.3 \text{ ps}) = 50 \text{ GHz}$), translational motions (around 0.5 THz) and librations (around 3 THz). As with DMSO, we observe enhancement of the TKE vs OKE signals when we shift the pump center frequency from 0.7 to 2 THz, i.e. toward the librational mode (see new Fig. 4b).

As seen in Fig. 4a vs 4b, this enhancement is even more pronounced than in DMSO which may be an indication of the process (ii) mentioned above. The reason is that in acetonitrile the frequency of the cage translational motion is closer to the libration frequency than in DMSO, thereby suggesting stronger (field-free) coupling between $\text{Lib}(P_2)$ and $\text{Reo}(P_2)$.

Action:

1) We added text (section "Discussion") as above to discuss the difference in the mode coupling of Fayer and colleagues and the THz-driven optical transition from P_1 -librations to P_2 -reorientational motion.

2) We added TKE/OKE data on acetonitrile to Fig. 4 (now Fig. 4b, DMSO is shown by Fig. 4a) and added text to explain and discuss the new acetonitrile data (section "Discussion").

To better illustrate the (off-) resonant transitions involved in optical and THz excitation, we added schematics (see new Figs. 5a,b).

In this context it would also be interesting to see the role of the molecular internal degrees of freedom on the response. The model systems chosen are very simple and the theoretical model matches this simplicity - this is not a criticism in the work presented, as it is very helpful indeed to prove the effect using a simple system, rather I wonder whether the authors might want to comment on this aspect. Have you studied larger molecules that exhibit greater intramolecular flexibility? I would expect even stronger differences between the THz and optical response and it would be interesting to see such results.

Response: We have not yet studied molecules with larger intramolecular flexibility. So far, we have investigated more than 20 liquids but did not observe signatures of intramolecular modes. As anticipated by the reviewer, this fact most likely arises because the liquids we observe are simple and consist of relatively small molecules that exhibit intramolecular vibrations at rather elevated frequencies above 3 THz.

Recently reported exceptions are heavy halogenated liquids, in which intramolecular modes were observed following intense THz excitation [please see Allodi et al. J. Chem. Phys. 143, 234204 (2015) & A. Finneran et al., PNAS 113 6857, (2016)]. We are currently pushing our intense THz sources toward higher frequencies which will enable us to excite the intramolecular modes of the liquids directly.

Action: We have added text to the introduction in which we refer to these works studying intramolecular contribution in the TKE response of liquids.

The paper briefly discusses the topic of radiation induced net heating effects in the experimental section. Have you investigated the role of transient heating at picosecond timescales on the measured birefringence dynamics? Clearly the magnitude of absorption of the EM wave is different at the two frequency ranges studied and it would be very useful to see a discussion of this effect.

Response: The temperature rise upon absorption of the THz pump pulse (for ~2 MV/cm pulses from our Lithium Niobate source) is less than 100 mK for the liquids studied here. In addition, heating effects are expected to cause only isotropic changes in the sample and thus does not contribute to the anisotropy signal, which is measured in our setup. Therefore, we do not expect a measureable effect of THz-induced sample heating.

This notion is corroborated by the following test experiment: when measuring TKE with and without liquid flow through the sample cell, no signal modification was observed within the noise level of the signal.

Action: We added text (section “Methods: Sample details”) to address this issue.

Please add the experimental details of the OKE setup used. The manuscript currently only describes the THz measurements.

Action: We regret this omission and added the OKE setup details to the section “Methods: Pump-probe setup”.

Caption Figure 2: remove 'to the birefringence' in line 3.

Action: Thanks a lot---we corrected accordingly.

Reviewers' comments:

Reviewer #1 (Remarks to the Author):

The revised manuscript by Sajadi et al. on the "Transient birefringence of liquids induced by terahertz electric-field torque on permanent molecular dipoles" has taken into account the great majority of the comments by the referees on the original submission to Nature Communications. In particular, for my own comments, I find the discussion of OKE vs. TKE responses considerably improved. And, the revised manuscript is certainly more up front with the previous work that has been done. I do have a few additional comments for the authors and editors to consider, on this revised version.

First, as to the science. Figure 5 presents a reasonable mechanism for the observed results, but also does raise a question or two. Most important, at room temperature, $h\nu/kT \sim 0.5$ for a 3 THz photon. Thus, the excited states outlined in this mechanism will have substantial population, and a ladder of such states can be envisioned that will interact with the THz and 800 nm fields. Might this have any impact on the results? Is there a strong temperature dependence predicted that might be tested with future observations (not required here)?

Second, I think some additional care is likely needed in the discussion of the μ_o and μ_{ind} components, and in the contribution of the permanent dipole moments and polarizabilities to the observed signatures. That is, the TKE signatures ultimately depend critically on the birefringence of the non-equilibrium liquid, and as such strong signatures do not require a permanent dipole moment. CS₂, for example, has a very strong TKE response, according to Hoffman et al. and Allodi et al. With librational modes in a liquid, there can be dipole FLUCTUATIONS at THz frequencies that drive absorption and that are present only in the liquid (that is, there are not present in gas phase samples), and that are responsible for the THz absorptivity of the material. These liquid-driven fluctuations are distinct from the light field-driven μ_{ind} discussed in this paper, and will be difficult to disentangle without theoretical work on both the isolated molecules and liquids. I do not suggest such work here, but do believe it should be noted in the discussion of the results.

Second, as to broader impacts. As noted above, previous work is now given better deference. But, I do believe some additional clarification would be helpful. Hoffman et al. did in fact study polar species - CH₂I₂ has a permanent dipole moment of >1 D. What they were not able to detect, and which is nicely done here, are STRONGLY polar liquids with significant microwave through THz absorption coefficients, and the nonlinear process that orients molecules is clearly discussed in this work and follow-on studies by the Nelson and Blake groups.

The work here is a nice step forward, but many groups have noted the possibility of coherent control. What these papers do NOT discuss, and what would be an extremely valuable addition here, in my view, would be to assess what THz fields are necessary for significant orientational control. That is, my sense of reading the literature is that all previous experiments, and likely this one as well, are in the perturbative regime, unlike, for example, NMR or very high field 2D-IR and -optical experiments. What is the estimated non-random orientational fraction in these experiments, and what would be needed for significant clustering of the permanent dipole orientations along the applied field polarization axis? Such an estimate, for a given species and permanent dipole moment, with the E_{THz} required, would greatly improve the broader impact of this work. Near the end of the paper is the natural location, which would also cite the rapidly improving prospects of fields in the >10 MV/cm regime.

A file with suggested grammatical and stylistic revisions has been sent to the editors.

Reviewer #2 (Remarks to the Author):

The responses to the referee comments adequately address all of the concerns, including in particular the comments concerning novelty. I therefore recommend in favor of publication.

Reviewer #3 (Remarks to the Author):

I would like to thank the authors for carefully addressing all the comments I made and sharing a detailed discussion of how they arrived at the conclusions they made in the manuscript. I am fully satisfied with the response given and I think the changes made during the revision of this manuscript have helped significantly to clarify the work and the novelty of the contribution. This manuscript reports a significant advance in the field and, whilst I can understand the initial reservations expressed by referee 1, in my view the work presented is novel and of broad interest and hence I strongly recommend publication of the revised manuscript.

Reviewer #1 (Remarks to the Author):

The revised manuscript by Sajadi et al. on the "Transient birefringence of liquids induced by terahertz electric-field torque on permanent molecular dipoles" has taken into account the great majority of the comments by the referees on the original submission to Nature Communications. In particular, for my own comments, I find the discussion of OKE vs. TKE responses considerably improved. And, the revised manuscript is certainly more up front with the previous work that has been done. I do have a few additional comments for the authors and editors to consider, on this revised version.

Response: We would like to thank the reviewer for her/his interest in and constructive and detailed comments on our work and her/his detailed and constructive comments in the first and second review round. We are very happy we could adequately address the previous points of the reviewer.

Our response to the reviewer's comments of the second review round is detailed point by point in the following. The resulting manuscript changes are highlighted yellow.

1) First, as to the science. Figure 5 presents a reasonable mechanism for the observed results, but also does raise a question or two. Most important, at room temperature, $h\nu/kT \sim 0.5$ for a 3 THz photon. Thus, the excited states outlined in this mechanism will have substantial population, and a ladder of such states can be envisioned that will interact with the THz and 800 nm fields. Might this have any impact on the results? Is there a strong temperature dependence predicted that might be tested with future observations (not required here)?

Response: We fully agree with the reviewer that the THz Kerr effect (TKE) can strongly depend on the thermal population of the states involved. Based on Fig. 5, we expect the major effect of thermal excitation to arise from the reduced population of the P_0 -like "ground state" which should result in a proportionally reduced transfer of population from P_0 to P_2 that is measured by the optical probe. In principle, two interactions with the pump field can also involve the thermal populations via other pump-interaction sequences such as $P_2 \rightarrow P_3 \rightarrow P_2$.

To qualitatively explore the impact of such effects in the framework of our model, we consider Eq(3) resulting from our simple modeling. This relationship indicates that a temperature dependence of the TKE can arise from the temperature dependence of χ^{μ_0} (the contribution of the permanent electric dipole moment μ_0 to the familiar total dielectric susceptibility of the liquid) and of R_2 (the response function describing the temporal relaxation of a P_2 -type perturbation of the angular distribution function of the molecules). Therefore, since both the GHz and THz dielectric susceptibility and the optical Kerr response of liquids may in general exhibit significant temperature dependence, we expect the same for the TKE signal, both in terms in amplitude and shape of the initial decay. For example, Debye-modes are known to shift their spectral peak when temperature is increased.

Experimentally, these effects can be studied by conducting TKE and OKE measurements and THz dielectric spectroscopy as a function of temperature. In fact, we are currently extending our setups to be able to systematically vary this important parameter.

Action: We added a paragraph right before the conclusion section: "Figure 5a suggests that thermal population of the various states may have a significant impact on the TKE signal. Indeed, since the THz dielectric susceptibility and the OKE signal of liquids can significantly depend on temperature, Eq. (3) implies the TKE signal does also. Therefore, varying the sample temperature in addition to the pump frequency is likely to deliver important information on the nature of the driven modes."

2) Second, I think some additional care is likely needed in the discussion of the μ_0 and μ_{ind} components, and in the contribution of the permanent dipole moments and polarizabilities to the observed signatures. That is, the TKE signatures ultimately depend critically on the birefringence of the non-equilibrium liquid, and as such strong signatures do not require a permanent dipole moment. CS₂, for example, has a very strong TKE response, according to Hoffman et al. and Allodi et al. With librational modes in a liquid, there can be dipole FLUCTUATIONS at THz frequencies that drive absorption and that are present only in the liquid (that is, there are not present in gas phase samples), and that are responsible for the THz absorptivity of the material. These liquid-driven

fluctuations are distinct from the light field-driven μ_{ind} discussed in this paper, and will be difficult to disentangle without theoretical work on both the isolated molecules and liquids. I do not suggest such work here, but do believe it should be noted in the discussion of the results.

Response: We fully agree with the reviewer that THz absorption by librations can (through the fluctuation-dissipation theorem) be understood as to arise from THz fluctuations of permanent molecular dipole moments (even when the molecule is nonpolar in the gaseous phase). This is not possible in gases where librational modes do not exist.

As of now, our TKE vs OKE data on nonpolar liquids render the contribution of THz absorption by such fluctuating dipoles (which can also arise from collisions) to the TKE signal rather small. To resolve such small effects, we may indeed need a higher signal-to-noise ratio and sophisticated theory support.

Action: We have extended the discussion part on THz absorption by dipole fluctuations in the section “Nonpolar vs polar liquids”. We have added a remark to the conclusion section: “Finally, more sophisticated models will help reveal the role of interaction-induced fluctuating dipoles in the solvent dynamics.”

3) Second, as to broader impacts. As noted above, previous work is now given better deference. But, I do believe some additional clarification would be helpful. Hoffman et al. did in fact study polar species - CH₂I₂ has a permanent dipole moment of >1 D. What they were not able to detect, and which is nicely done here, are STRONGLY polar liquids with significant microwave through THz absorption coefficients, and the nonlinear process that orients molecules is clearly discussed in this work and follow-on studies by the Nelson and Blake groups.

Response: We thank the reviewer for this clarification.

Action: We added text at various locations (e.g. abstract, introduction, conclusion) to emphasize that solvent molecules considered here are strongly polar, in contrast to Hoffmann et al.

4) The work here is a nice step forward, but many groups have noted the possibility of coherent control. What these papers do NOT discuss, and what would be an extremely valuable addition here, in my view, would be to assess what THz fields are necessary for significant orientational control. That is, my sense of reading the literature is that all previous experiments, and likely this one as well, are in the perturbative regime, unlike, for example, NMR or very high field 2D-IR and -optical experiments. What is the estimated non-random orientational fraction in these experiments, and what would be needed for significant clustering of the permanent dipole orientations along the applied field polarization axis? Such an estimate, for a given species and permanent dipole moment, with the E_{THz} required, would greatly improve the broader impact of this work. Near the end of the paper is the natural location, which would also cite the rapidly improving prospects of fields in the >10 MV/cm regime.

Response: We thank the reviewer for this excellent suggestion. For broader impact, we should indeed make an estimate of the expected molecular alignment that can be induced with existing (or possibly soon available) high-field THz sources.

Action: We accordingly added text to the conclusion section, extended the Methods section “Pump-probe setup” and added a new Methods section “Alignment estimate”.

A file with suggested grammatical and stylistic revisions has been sent to the editors.

Response: We again would like to thank the reviewer for her/his very detailed efforts and comments. They have led to a significantly improved manuscript which is now (we believe) suitable for publication in Nature Communications.

REVIEWERS' COMMENTS:

Reviewer #1 (Remarks to the Author):

The authors have done a very nice job incorporating the last round of comments on this manuscript. I have no further detailed comments. Below I list two simple grammatical corrections from the highlighted (in yellow) changes:

Line 217: Spell out verus

Line 345: In the case of negligible velocity mismatch between the pump and probe pulses, the birefringence-induced phase shift experienced by the probe is given by ...

Reviewer #1 (Remarks to the Author):

The authors have done a very nice job incorporating the last round of comments on this manuscript. I have no further detailed comments.

Response: We would like to thank the reviewer for her/his constructive and detailed comments on our work and her/his constructive comments in the previous review rounds.

Below I list two simple grammatical corrections from the highlighted (in yellow) changes:

Line 217: Spell out versus

Line 345: In the case of negligible velocity mismatch between the pump and probe pulses, the birefringence-induced phase shift experienced by the probe is given by ...

Action: Thank you very much. They were corrected accordingly.